# VLX600, an anticancer iron chelator, exerts antimicrobial effects on *Mycobacterium abscessus* infections

Jaehun Oh,[1,2] Seaone Choi,[1,2] Hyejun Seo,[1,3] Dong Hyun Kim,[1,2] Hyelin Kim,[1,3] Duhyung Lee,[1,3] Junghwa Jang,[1,2] Sangkwon Jung,[1,3] Ju-Young Lee,[1,2] Ziyun Kim,[1,2] Jae-Joon Yim,[4,5] Nakwon Kwak,[4,5] Bum-Joon Kim[1,2,3,6,7]

**ABSTRACT** *Mycobacterium abscessus* presents significant clinical challenges due to its intrinsic and acquired resistance to antibiotics, resulting in prolonged treatments and poor patient outcomes. Addressing the urgent need for novel therapeutics, this study explores the antimicrobial potential of VLX600, originally developed as an anticancer agent, against *M. abscessus*. Screening a library of 3,200 clinically evaluated compounds identified VLX600 as a potent antimicrobial with minimal cytotoxicity. VLX600 demonstrated inhibitory effects against various strains of *M. abscessus* with minimum inhibitory concentrations of 4 µg/mL–16 µg/mL. It also remained effective in intracellular *M. abscessus* in host cells and exhibited broad-spectrum activity against other bacterial species, including *Escherichia coli*, *Staphylococcus aureus*, and *Pseudomonas aeruginosa*. The antimicrobial activity of VLX600 was abrogated by supplemental iron, indicating a mechanism dependent on iron chelation. VLX600 significantly reduced bacterial burdens and inflammation in a murine model of pulmonary *M. abscessus* infection. Additionally, synergistic effects were observed when VLX600 was combined with conventional antibiotics such as amikacin and clarithromycin *in vitro*. These findings highlight VLX600 as a promising candidate for repurposing as an antimicrobial agent against *M. abscessus*, warranting further clinical investigations.

**IMPORTANCE** *Mycobacterium abscessus* is an opportunistic pathogen that commonly causes pulmonary infections in cystic fibrosis patients. These infections are notoriously difficult to treat due to high levels of antibiotic resistance of *M. abscessus*, resulting in low cure rates. In this study, we identified a novel antibiotic candidate, VLX600, through high-throughput screening of 3,200 clinical compounds and demonstrated that VLX600 inhibits the growth of *M. abscessus* by depriving it of ferric and ferrous ions. This study highlights the potential of iron chelators as antimicrobial agents against *M. abscessus* infections. Since iron is an essential nutrient for the growth of many bacteria, the use of iron chelators could be extended to other infectious diseases. We hope this research will inspire further studies aimed at developing iron chelators as a novel class of antimicrobial agents.

**KEYWORDS** *Mycobacterium abscessus*, VLX600, drug repositioning, antibiotics, iron chelator

Nontuberculous mycobacteria (NTM) are increasingly recognized as significant opportunistic pathogens causing a variety of infections in humans (1, 2). Among NTM species, *Mycobacterium abscessus* is particularly problematic, being one of the most common causes of NTM pulmonary disease (3). Pulmonary infections caused by *M. abscessus* predominantly occur in individuals with underlying conditions such as cystic fibrosis (CF), chronic obstructive pulmonary disease, or bronchiectasis (4, 5). *M. abscessus* infections have emerged as a global public health concern due to their rising prevalence

Address correspondence to Bum-Joon Kim, kbumjoon@snu.ac.kr.

Jaehun Oh and Seaone Choi contributed equally to this article. Author order was determined on the basis of contribution and seniority.

J.-J. Yim has served as the overall or institutional principal investigator for clinical trials related to non-tuberculous mycobacterial pulmonary disease sponsored by LigaChem Biosciences, Insmed, and AN2 Therapeutics. Additionally, he has received several drugs free of charge as a principal investigator for previous trials related to tuberculosis from Pfizer, Otsuka, and Yuhan.

See the funding table on p. 15.

and transmission potential. Recent genomic analyses have revealed person-to-person transmission in CF patients, challenging the long-held notion that infections are solely acquired from environmental sources (6, 7).

*M. abscessus* is characterized by its rapid growth and inherent resistance to a wide range of antibiotics (8, 9), including first-line antitubercular agents and other common classes such as beta-lactams, quinolones, and macrolides (10). Consequently, treatment success rates for *M. abscessus* infections are alarmingly low, often ranging between 30% and 50% (11). Despite these challenges, there is no standard drug regimen for *M. abscessus*, and treatment remains prolonged, complex, and largely ineffective for many patients (12).

Given the urgent need for effective therapies, drug repositioning has emerged as a promising strategy (13). This approach significantly reduces the time and cost compared to traditional drug development. Among existing candidates, compounds targeting bacterial iron metabolism have gained attention as iron is an essential nutrient for bacterial survival and virulence. Strategies to inhibit iron acquisition have shown potential against pathogenic bacteria in previous studies (14–16).

In this research, we discovered VLX600 as a novel antimycobacterial agent for the treatment of *M. abscessus* infection through a high-throughput screening system using bioluminescent reporter gene. VLX600, initially developed as an anticancer agent that targets mitochondrial oxidative phosphorylation, has demonstrated anticancer efficacy by disrupting metabolic pathways and promoting autophagy in tumor cells (17).

Here, we report that VLX600 exhibits strong antimycobacterial efficacy by selectively depriving mycobacteria of iron and suggest that VLX600 is a potential treatment option for *M. abscessus* infections.

## RESULTS

### Construction of bioluminescent strains of *M. abscessus* and evaluation as reporters of bacterial viability

Recombinant *M. abscessus* strains emitting bioluminescence were engineered for the efficient high-throughput screening of a chemical library. These strains were constructed by introducing the pMV306hsp+luxG13 plasmid into three clinical isolates of *M. abscessus* (18) (Fig. S1A). The clinical isolates were acquired from patients with pulmonary M. abscessus infection at Seoul National University Hospital, and their identification numbers and genetic profiles are listed in Table S1. The plasmid was introduced through electroporation, and the luminescence of the strains was detected using a Tecan F200 microplate reader (Fig. S1B). All bioluminescent strains of *M. abscessus* presented a positive and linear correlation between the luminescence intensity and optical density at 600 nm ($OD_{600}$) (Fig. S1C). In addition, the luminescence of these strains was closely related to their viability, as evidenced by the addition of amikacin, which decreased the luminescence in proportion to its concentration (Fig. S1D). The luminescence of the bacteria was maintained in a J774A.1 infection model, and the intensity increased depending on the multiplicity of infection (MOI) (Fig. S1E).

These findings validate the use of a luminescence-based reporter system for screening antimycobacterial compounds.

### Antimycobacterial screening with bioluminescent *M. abscessus* strains

A randomized chemical drug library comprising 3,200 compounds that have been tested in a clinical setting was provided by the Korea Chemical Bank and screened for their antimycobacterial activity against *M. abscessus* under dual-blind conditions. The antimycobacterial activity of the chemical library was evaluated using the bioluminescent *M. abscessus* rough morphotype strain (ID number 5) at a single-dose treatment of 5 µM. Given that most NTMs are intracellular pathogens, the initial screen was performed using two approaches: a direct treatment model and a J774A.1 infection model (Fig. S2). A detailed screening procedure is described in the supplemental material (Fig.

S3). Cytotoxicity of selected compounds was evaluated using the neutral red uptake (NRU) assay, and two compounds, with negligible cytotoxicity, VLX600 and 226-C4, were ultimately identified as hit compounds (Fig. 1A). In this report, the antimycobacterial potential of VLX600, one of the two remaining compounds, was investigated. The selected hit compound, VLX600, exhibited significant antimycobacterial activity against bioluminescent *M. abscessus* under direct treatment conditions, whereas clarithromycin displayed poor activity due to the presence of the *erm* (41) gene (Fig. 1B). Moreover, VLX600 remained effective against intracellular bioluminescent *M. abscessus* when infected J774A.1 macrophages were treated with VLX600 (Fig. 1C).

## VLX600 is an effective bacteriostatic agent that inhibits both the extracellular and intracellular growth of *M. abscessus* in macrophages

Conventional assays, such as the broth microdilution assay and colony forming unit (CFU) assay, rather than the bioluminescence-based system, were used to evaluate the antimycobacterial activity of VLX600 for validation. First, the minimum inhibitory concentrations (MICs) of VLX600 were determined against various strains of *M. abscessus* according to Clinical and Laboratory Standards Institute (CLSI) guidelines using cation-adjusted Mueller-Hinton broth (CAMHB) (19, 20). The MICs of VLX600 determined against all type strains of *M. abscessus* subsp. *abscessus*, *massiliense*, and *bolletii* were 8 µg/mL (Fig. 2A). The MICs of VLX600 against clinical isolates of *M. abscessus* subsp. *abscessus* ranged from 4 µg/mL to 16 µg/mL, and the MICs against clinical isolates of *M. abscessus* subsp. *massiliense* ranged from 8 µg/mL to 16 µg/mL, indicating the sensitivity of *M. abscessus* to VLX600 is highly strain-specific. The MICs of VLX600 against type strains and clinically isolated strains are shown in Table 1. The growth inhibitory effect of VLX600 was also evaluated with a CFU assay in a growth culture. A significant reduction in CFU was observed, depending on the concentration of VLX600 treated (Fig. 2B).

The minimum bactericidal concentrations (MBCs) of VLX600 against each of the three type strains of *M. abscessus* were also determined following the MIC determination to elucidate its bacteriostatic or bactericidal profile. All the MBC values determined for each of the three type strains of *M. abscessus* were 64 µg/mL, which exceeded a fourfold ratio relative to the MIC, suggesting that VLX600 acts as a bacteriostatic agent against *M. abscessus* rather than a bactericidal agent (21). The measured MBCs against the type strains are presented in Table 2.

The MICs of VLX600 against other *Mycobacterium* species and other genera were also determined to assess the antimicrobial spectrum of VLX600. Eleven species of *Mycobacterium*, including *Mycobacterium avium*, *Mycobacterium intracellulare* subsp. *yongonense*, *Mycobacterium paraintracellulare*, *Mycobacterium bovis* BCG, *Mycobacterium gordonae*, *Mycobacterium paragordonae*, *Mycobacterium marinum*, *Mycobacterium vaccae*, *Mycobacterium terrae*, *Mycobacterium chleonae* subsp. *bovistauri*, and *Mycobacterium chelonae* subsp. *gwanakae*, were utilized for the MIC test. For the assessment of other genera, *Escherichia coli*, *Staphylococcus aureus*, and *Pseudomonas aeruginosa* were utilized for the MIC test. VLX600 effectively inhibited the growth of other mycobacterial species, with MIC values ranging from 1 µg/mL to 8 µg/mL (Fig. S4), and the MIC values against *Mycobacterium* spp. are presented in Table 3. VLX600 was also effective at inhibiting the growth of *E. coli*, *S. aureus*, and *P. aeruginosa*, with MIC values of 16, 16, and 4 µg/mL, respectively (Fig. S5). These results indicate the broad-spectrum activity of VLX600 as an antimicrobial agent.

The antimycobacterial activity of VLX600 was subsequently evaluated in the murine macrophage line J774A.1 infected with *M. abscessus* by treating the infected cells with various concentrations of VLX600. Compared with the vehicle control, treatment with VLX600 resulted in a dose-dependent reduction in CFUs in macrophages (Fig. 2C).

**(A)**

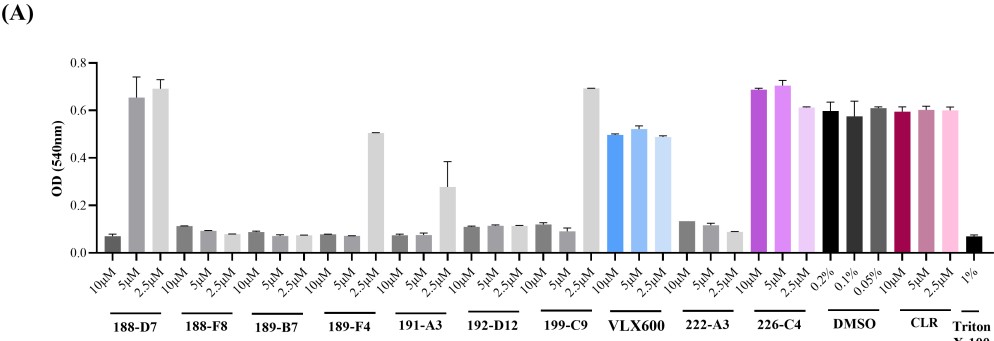

**(B)**

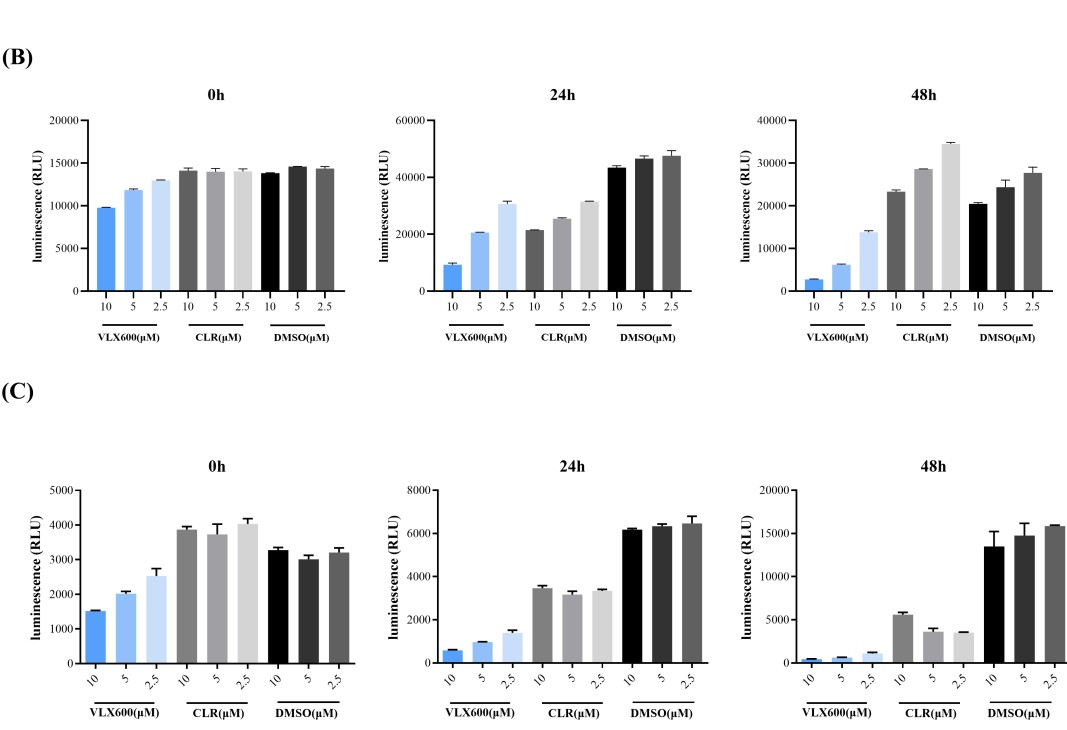

**FIG 1** Screening result from the chemical library and discovery of VLX600 as a hit compound. (A) An NRU assay was performed at 48 h after drug treatment to confirm the cytotoxicity of different doses of the 10 selected compounds on J774A.1 cells ($n = 3$). The samples were treated with 1% Triton X-100 for 10 min before staining and used as a positive control. (B) Antimycobacterial activity of VLX600 in the direct treatment model. VLX600 was applied to the bioluminescent *M. abscessus* cultures, and the luminescence was measured every 24 h ($n = 3$). (C) Antimycobacterial activity of VLX600 in the J774A.1 infection model. J774A.1 cells were infected with bioluminescent *M. abscessus* at an MOI of 10, followed by treatment with VLX600 at different doses ($n = 3$). Luminescence was measured every 24 h. The error bars represent the standard errors of the means.

## VLX600 inhibits the growth of *M. abscessus* through its specific iron-chelating activity

VLX600 has been reported to exert anticancer effects through the inhibition of mitochondrial respiration by chelating ferric and ferrous ions, and the anticancer effect of VLX600 was abrogated by the addition of these ions to the cell culture (22). Iron is an irreplaceable essential nutrient for most bacteria, especially in environments where access to free iron is extremely limited, such as during infection (23, 24). Therefore, iron chelation has been a potential strategy for developing novel antibiotics (25). In this context, the relationship between the iron-chelating ability and antimicrobial activity of VLX600 was investigated. As a result, the antimicrobial activity of VLX600 against *M. abscessus* was critically abrogated by the addition of $FeCl_2$ and $FeCl_3$ in a dose-dependent manner, whereas it remained effective in the presence of other metal ions with the

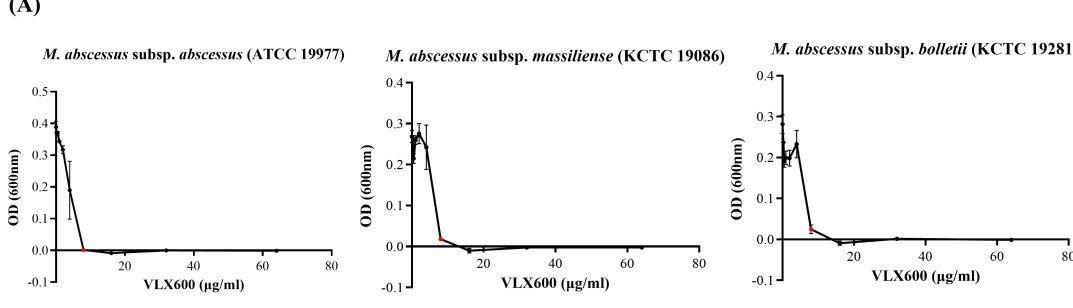

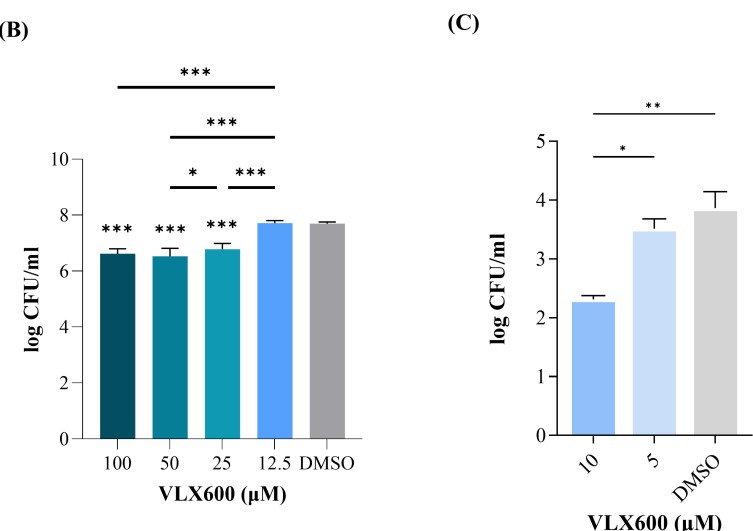

**FIG 2** Antimycobacterial efficacy of VLX600 against type strains of *M. abscessus*. (A) The MICs of VLX600 against type strains of *M. abscessus* were determined with a broth microdilution assay. Each strain of *M. abscessus* was diluted in CAMHB to McFarland standard 0.5 and further diluted 1:100 in CAMHB. VLX600 was serially diluted in CAMHB and applied to *M. abscessus* cultures in 96-well plates (*n* = 3). The plates were then cultured at 37°C for 1 week, and the $OD_{600}$ was measured to determine the MIC. The MIC was defined as the lowest concentration of VLX600 that inhibited bacterial growth by at least 90%. The MICs are shown as red dots on the graphs. (B) The antimycobacterial efficacy of VLX600 against *M. abscessus* was evaluated with a CFU assay in the direct treatment model. The type strain of *M. abscessus* subsp. *abscessus* (ATCC 19977) culture from the broth microdilution assay was serially diluted in phosphate-buffered saline (PBS) and spread on 7H10 agar plates. The plates were incubated at 37°C until colonies appeared. Asterisks above the bars indicate statistical significance compared to the vehicle. (C) . The antimycobacterial efficacy of VLX600 in the J774A.1 infection model was evaluated with a CFU assay. J774A.1 cells were infected with the type strain of *M. abscessus* subsp. *abscessus* (ATCC 19977) at an MOI of 10, followed by treatment with different doses of VLX600 (*n* = 3). The plates were then cultured at 37°C for 48 h. The intracellular bacteria were harvested by lysing the cells with 1% Triton X-100, and then samples were spread on 7H10 agar plates for the CFU assay. The statistical significance was determined by one-way analysis of variance with Tukey's multiple comparison test, and the results are denoted as follows: *$P < 0.05$, **$P < 0.01$, and ***$P < 0.001$. The error bars represent the standard errors of the means.

exception of CuCl (Fig. 3). Our results indicate that VLX600 inhibits the growth of *M. abscessus* by chelating $Fe^{2+}$ and $Fe^{3+}$ ions, thereby depriving the bacteria of iron.

## VLX600 exerts synergistic effects in combination with amikacin and clarithromycin on inhibiting the growth of *M. abscessus*

The treatment of *M. abscessus* infections commonly involves a combination of multiple antibiotics to overcome its high level of antibiotic resistance (26, 27). Therefore, the potential synergistic effects of a novel drug and conventional antibiotics could be advantageous for the development of drugs that target *M. abscessus* infections. As such, the synergistic effects of VLX600 in combination with amikacin and clarithromycin, which are commonly used to treat *M. abscessus* infection, were investigated using a

**TABLE 1** MICs of VLX600 against type strains and clinical isolates of *M. abscessus*

| *M. abscessus* strain or isolate | MIC (µg/mL) |
| --- | --- |
| Type strains | |
| *M. abscessus* subsp. *abscessus* | 16 |
| *M. abscessus* subsp. *massiliense* | 8 |
| *M. abscessus* subsp. *bolletii* | 8 |
| Clinical isolates | |
| *M. abscessus* subsp. *abscessus* 5 | 8 |
| *M. abscessus* subsp. *abscessus* 8 | 4 |
| *M. abscessus* subsp. *abscessus* 11 | 4 |
| *M. abscessus* subsp. *abscessus* 16 | 8 |
| *M. abscessus* subsp. *abscessus* 19 | 8 |
| *M. abscessus* subsp. *abscessus* 23 | 16 |
| *M. abscessus* subsp. *abscessus* 28 | 8 |
| *M. abscessus* subsp. *abscessus* 37 | 16 |
| *M. abscessus* subsp. *abscessus* 40 | 16 |
| *M. abscessus* subsp. *massiliense* 26 | 16 |
| *M. abscessus* subsp. *massiliense* A56 | 8 |
| *M. abscessus* subsp. *massiliense* B115 | 16 |
| *M. abscessus* subsp. *massiliense* B152 | 8 |
| *M. abscessus* subsp. *massiliense* C11 | 16 |
| *M. abscessus* subsp. *massiliense* E22 | 8 |
| *M. abscessus* subsp. *massiliense* E50 | 8 |
| *M. abscessus* subsp. *massiliense* G53 | 8 |
| *M. abscessus* subsp. *massiliense* H05 | 8 |

checkerboard assay. Synergy was evaluated with two approaches: a direct treatment model and a J774A.1 infection model.

Synergy was defined as an fractional inhibitory concentration index (FICI) value of 0.5 or lower (28). A strong synergistic effect was observed between VLX600 and clarithromycin, as well as between VLX600 and amikacin, against the *M. abscessus* subsp. *abscessus* type strain. However, for the *M. abscessus* subsp. *massiliense* type strain, a synergistic effect was observed between only VLX600 and clarithromycin. In the case of *M. abscessus* subsp. *bolletii*, only an indifferent action between VLX600 and these antibiotics was observed (Fig. 4A).

Synergy was evaluated by calculating the inhibition rate with four reference models, zero interaction potency (ZIP), Bliss, Loewe, and highest single agent (HSA), respectively, using the SynergyFinder web application. Synergy was defined as a synergy score greater than 5 (29, 30). An intensive synergistic effect was observed between VLX600 and clarithromycin on the J774A.1 infection model, as all the synergy scores exceeded 5. A synergistic effect was also observed between VLX600 and amikacin, as calculated with the ZIP and Bliss models, although varying results were observed with the Loewe and HSA models (Fig. 4B). Collectively, these results demonstrate that VLX600 strongly synergizes with amikacin and clarithromycin to inhibit the growth of *M. abscessus*.

**TABLE 2** MICs and MBCs of VLX600 against *M. abscessus* type strains

| Subspecies | MIC (µg/mL) | MBC (µg/mL) | MBC/MIC ratio | Profile |
| --- | --- | --- | --- | --- |
| *abscessus* | 8 | 64 | 8 | Bacteriostatic |
| *massiliense* | 8 | 64 | 8 | Bacteriostatic |
| *bolletii* | 8 | 64 | 8 | Bacteriostatic |

**TABLE 3** MICs of VLX600 against *Mycobacterium* spp

| Species | Strain | MIC (µg/mL) |
|---|---|---|
| *M. avium* subsp. *avium* | ATCC 25291 | 4 |
| *M. bovis* BCG | BCG Tokyo | 4 |
| *M. chelonae* subsp. *bovistauri* | QIA-37 | 4 |
| *M. chelonae* subsp. *gwanakae* | MOTT36W | 4 |
| *M. gordonae* | ATCC 14470 | 8 |
| *M. intracellulare* subsp. *yongonense* | 05-1390$^T$ | 4 |
| *M. marinum* | ATCC 927 | 4 |
| *M. paragordonae* | 49061$^T$ | 8 |
| *M. paraintracellulare* | MOTT64 | 4 |
| *M. terrae* | ATCC 15755 | 1 |
| *M. vaccae* | ATCC 15483 | 4 |

## VLX600 effectively reduces the *M. abscessus* burden in a mouse pulmonary infection model

A mouse pulmonary infection model using cyclophosphamide was utilized to explore the efficacy of VLX600 against *M. abscessus* infection *in vivo* (31, 32). The mice were rendered neutropenic by the intraperitoneal administration of 150 mg/kg cyclophosphamide on days 1 and 4 before infection. The mice were subsequently challenged with 1 × 10$^6$ CFUs of a clinical isolate of *M. abscessus* (ID number 5) through intranasal injection following anesthesia on day 0. To assess the efficacy of the model, three mice were sacrificed on day 1 post-infection, and lung CFUs were compared to those of mice that had not received cyclophosphamide (Fig. S6). Subsequently, the remaining infected mice were randomly divided into five groups, and treatment began the day after infection, consisting of daily intraperitoneal administration of PBS (vehicle), 5 mg/kg VLX600, 10 mg/kg VLX600, 50 mg/kg amikacin, or a combination of 50 mg/kg amikacin and 5 mg/kg VLX600. The VLX600 doses were determined based on the maximum tolerated dose and twice the dose reported in previous literature (17). The weights of the mice were monitored throughout the experiment. The mice were euthanized on day 12 after infection for further analysis (Fig. 5A).

Whereas the PBS group experienced substantial weight loss following infection, all treated groups maintained their body weight (Fig. 5B). Although no statistical significance was observed among these groups, except for the PBS group, the group that received the combination of VLX600 and amikacin showed the greatest weight gain, distinguishing it from the others. Consistently, CFU analysis of lung homogenates revealed that both VLX600 and amikacin significantly reduced the bacterial burden in the lungs compared to the PBS group (Fig. 5C; Table S2). In line with the CFU result, spleen enlargement was also significantly reduced in all treated groups, further indicating a reduction in bacterial burden (Fig. 5D). However, spleen enlargement in the 10 mg/kg VLX600 group was slightly greater than the 5 mg/kg VLX600 group, which was inconsistent with the CFU results. Moreover, the histological analysis of the lung through hematoxylin and eosin (H&E) staining revealed that the lung inflammation caused by infection was notably relieved in all treated groups compared to the PBS group (Fig. 5E). In summary, these results demonstrate that VLX600 has potential in reducing bacterial burden and alleviating lung inflammation in *M. abscessus* infection *in vivo*, whereas the additional benefit of combining it with amikacin remains uncertain.

## DISCUSSION

*M. abscessus* infections pose a significant clinical challenge due to their intrinsic resistance to most antibiotics (8–10). This resistance highlights the urgent need for new therapeutic strategies. However, the development pipeline for novel drugs against *M. abscessus* remains sparse, underscoring the importance of innovative approaches to combat this pathogen (33).

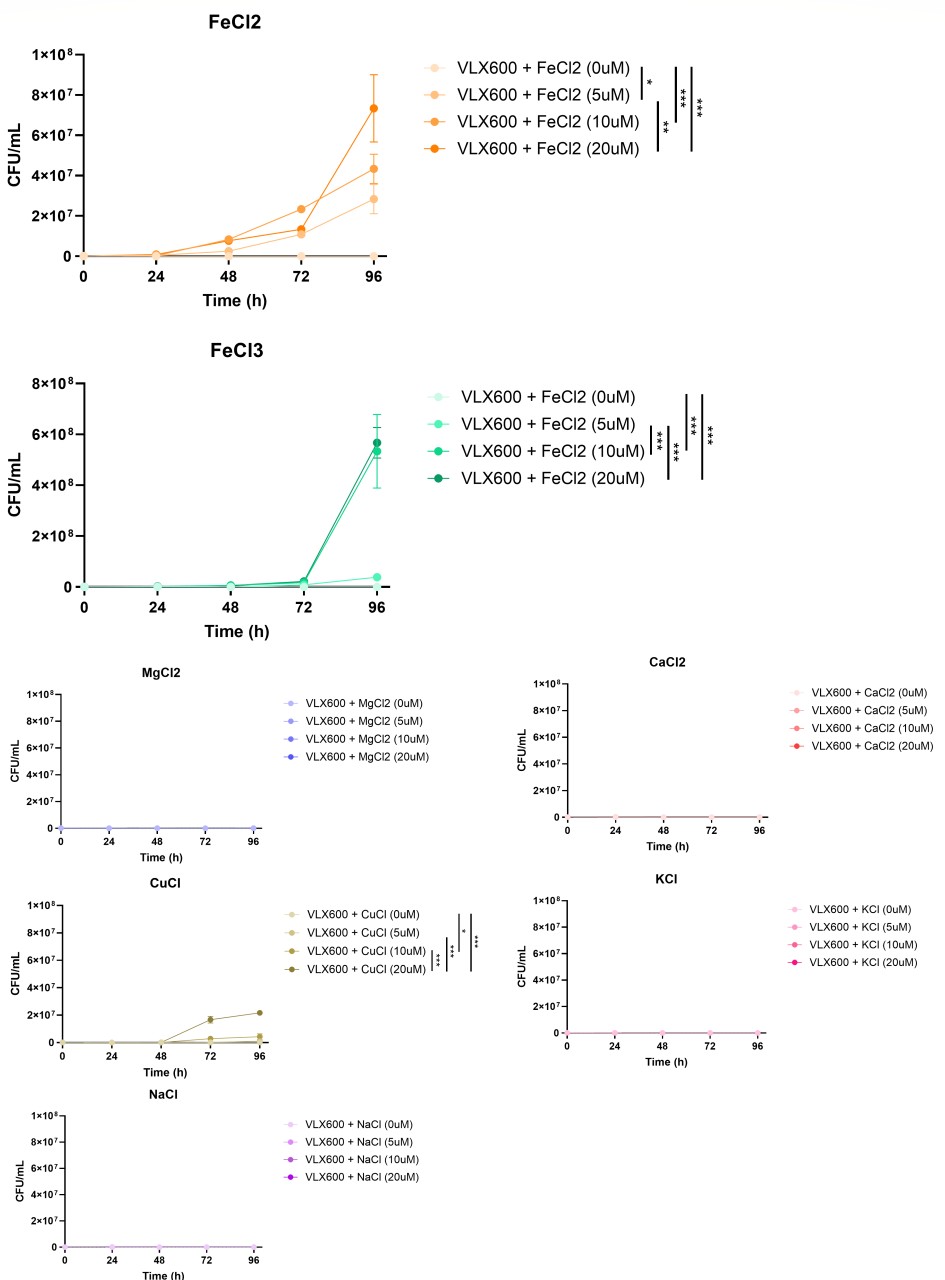

**FIG 3** Abrogation of the antimycobacterial activity of VLX600 by the addition of iron. The antimycobacterial activity of VLX600 (16 µg/mL) against the type strain of *M. abscessus* subsp. *abscessus* (ATCC 19977) was evaluated in the presence of various metal ions. The cultures were incubated at 37°C, and the CFUs were measured daily (*n* = 3). The antimycobacterial activity of VLX600 was only abrogated by the addition of $Fe^{2+}$ and $Fe^{3+}$ in a dose-dependent manner, whereas the addition of other metal ions did not affect its activity except for $Cu^+$. The statistical significance was determined by two-way analysis of variance with Tukey's multiple comparison test, and the results are denoted as follows: *$P < 0.05$, **$P < 0.01$, and ***$P < 0.001$. The error bars represent the standard errors of the means.

In this study, we identified VLX600, a novel antimicrobial agent, through a high-throughput screen of a chemical library comprising 3,200 clinical drugs. VLX600 demonstrated potent activity against *M. abscessus*, with MIC values ranging from 4 to 16 µg/mL (Table 1), and effectively reduced intracellular bacterial loads in macrophages (Fig. 1C). This activity was shown to be dependent on iron chelation, as its efficacy was

**(A)**

| | *subsp.* | Clarithromycin | Amikacin |
|---|---|---|---|
| **Type strains** | *M. abscessus* | 0.141 | 0.266 |
| | *M. massiliense* | 0.133 | 0.516 |
| | *M. bolletii* | 0.75 | 0.75 |

| | |
|---|---|
| FICI ≤ 0.5 | Synergy |
| 0.5 < FICI ≤ 4 | Indifferent |
| FICI > 4 | Antagonism |

**(B)**

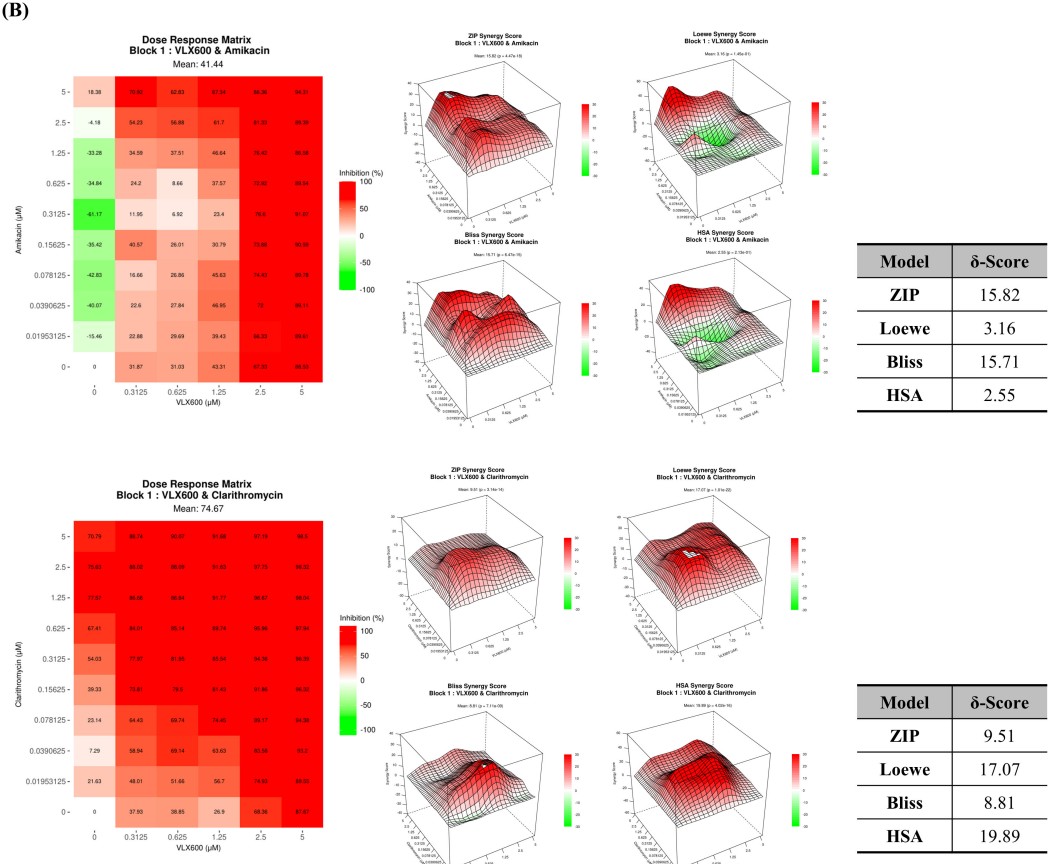

| Model | δ-Score |
|---|---|
| **ZIP** | 15.82 |
| **Loewe** | 3.16 |
| **Bliss** | 15.71 |
| **HSA** | 2.55 |

| Model | δ-Score |
|---|---|
| **ZIP** | 9.51 |
| **Loewe** | 17.07 |
| **Bliss** | 8.81 |
| **HSA** | 19.89 |

**FIG 4** Synergistic effects of VLX600 with amikacin and clarithromycin on *M. abscessus*. (A) The synergistic effect of VLX600 was evaluated on a direct treatment model using a checkerboard assay. The FICI was calculated, and the drug interaction was considered synergistic for FICI ≤ 0.5, indifferent for 0.5 < FICI ≤ 4, and antagonistic for FICI > 4. (B) The synergistic effect of VLX600 was evaluated on a J774A.1 infection model using a checkerboard assay. The luminescence was measured at 24 h post-infection, and the inhibition rate was calculated from the luminescence value. The drug interaction was evaluated using SynergyFinder with four methods: zero interaction potency (ZIP), Loewe, Bliss, and highest single agent (HSA). The drug interaction was considered synergistic when the score was ≥5, indifferent when the score was −5 < score < 5, and antagonistic when the score was ≤−5.

abolished by the addition of $Fe^{2+}$ and $Fe^{3+}$ ions, while other metal ions had no effect except for $Cu^+$ (Fig. 3). Although VLX600 was previously reported not to interact with $Cu^{2+}$ (22), a recent comprehensive study demonstrated that VLX600 forms complexes with both $Cu^+$ and $Cu^{2+}$, which explains the impact of CuCl on VLX600 activity in our study (34). The addition of $Cu^+$ might interfere with the iron-chelating activity of VLX600. These findings suggest that VLX600 acts by chelating iron, depriving *M. abscessus* of this essential nutrient and thereby inhibiting its growth.

Iron is critical for bacterial survival, particularly in the iron-restricted environment of the human host (35, 36). Pathogens like mycobacteria have evolved sophisticated mechanisms to scavenge iron, such as producing siderophores like mycobactin and exochelin. Mycobactin is a cell wall-associated siderophore, while exochelin is secreted

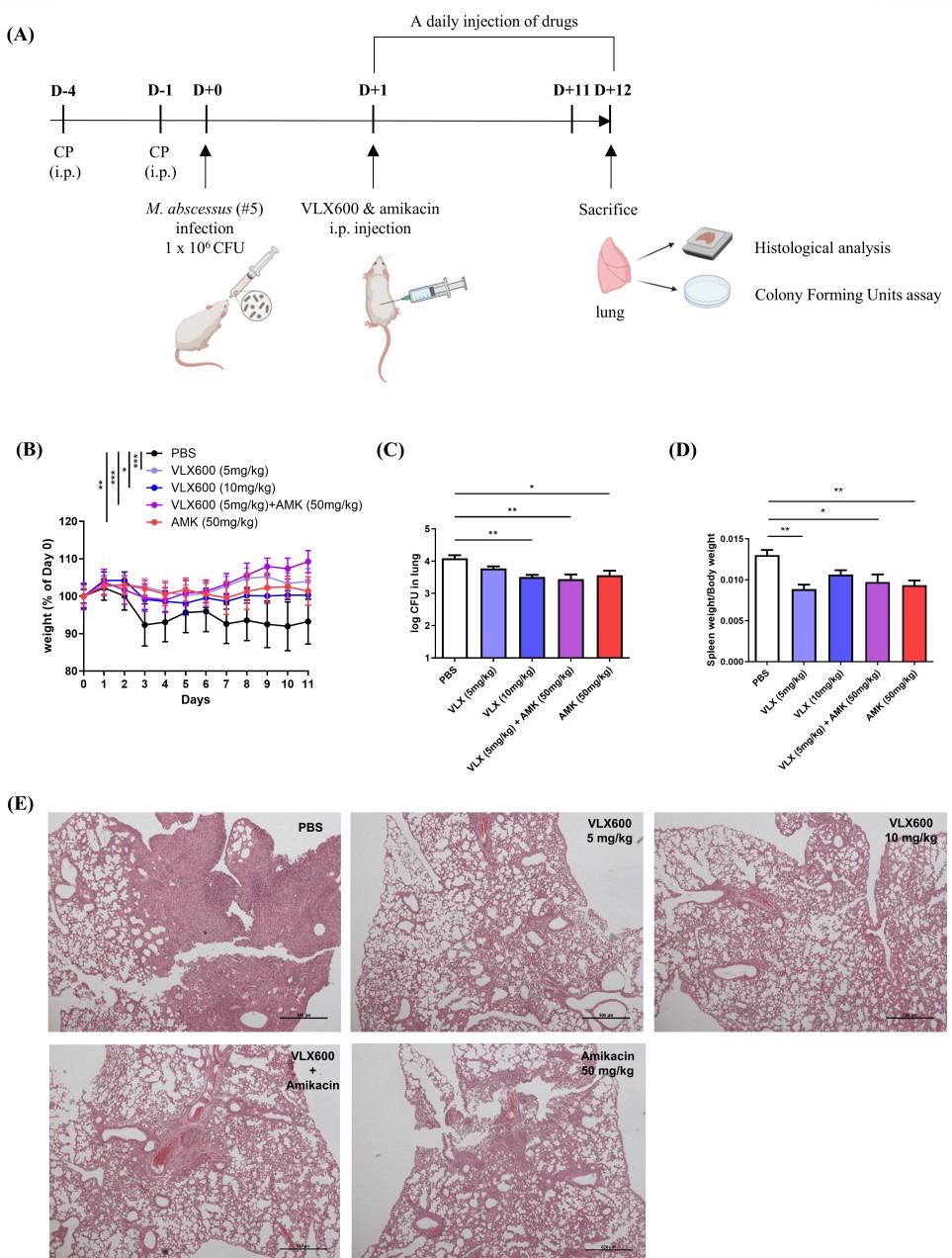

**FIG 5** *In vivo* efficacy of VLX600 in a mouse model of pulmonary infection (A) Schematic of the schedule of the *in vivo* experiment. The mice were intraperitoneally injected with 150 mg/kg cyclophosphamide on days 1 and 4 before infection to induce a neutropenic state. On day 0, the mice were challenged with $1 \times 10^6$ CFUs of a clinical isolate of *M. abscessus* via the intranasal route. The drug treatments started on day 1 and were administered daily until sacrifice on day 12 (*n* = 5). (B) The body weights of the mice were measured daily until sacrifice, and the results revealed a significant improvement in weight loss in all treated groups compared to the PBS group. (C) Lungs were excised and homogenized in PBS on the day of sacrifice. The homogenates were serially diluted and spread on 7H10 agar plates for CFU assays. (D) Spleens were excised and weighed to assess spleen enlargement in mice. The weight of each spleen was normalized to the corresponding body weights. (E) The postcaval lobe of each lung was fixed and stained with hematoxylin and eosin. The paraffin sections were subsequently observed under a light microscope as sectioned paraffin block. The statistical significance was determined by one-way analysis of variance (ANOVA) or two-way ANOVA with Tukey's multiple comparison test, and the results are denoted as follows: $*P < 0.05$, $**P < 0.01$, and $***P < 0.001$. The error bars represent the standard errors of the means.

into the extracellular environment, facilitating iron acquisition from host iron-binding proteins like transferrin and lactoferrin (37). The importance of siderophores in virulence is well-documented; for instance, deficiencies in siderophore biosynthesis and export significantly attenuate *Mycobacterium tuberculosis* in infection models (38). Given this, targeting iron acquisition with iron chelators like VLX600 is a compelling antimicrobial strategy (39, 40).

The combinatorial nature of *M. abscessus* treatment regimens makes drug synergy an invaluable property. VLX600 exhibited strong synergistic effects with amikacin and clarithromycin, two cornerstone antibiotics in *M. abscessus* treatment (Fig. 4). This synergy not only enhances antibacterial efficacy but also suggests the potential for reduced dosages of individual drugs, which could minimize side effects and resistance development.

The *in vivo* efficacy of VLX600 was demonstrated in a mouse pulmonary infection model (Fig. 5). VLX600 significantly reduced bacterial burden and inflammation in the lungs, achieving effects comparable to amikacin at a lower dosage. Splenomegaly, a marker of systemic infection, was alleviated in VLX600-treated groups, further supporting its efficacy. However, the slight spleen enlargement in the 10 mg/kg VLX600 group compared to the 5 mg/kg group suggests a potential link to iron-deficiency anemia (IDA). IDA has been associated with splenomegaly due to compensatory extramedullary hematopoiesis, a phenomenon observed in both clinical and experimental studies (41, 42). Given that anemia has been frequently reported in phase I clinical trials of VLX600, the slight spleen enlargement observed may support this hypothesis and warrant careful consideration of iron-related side effects in clinical use (43).

VLX600 also showed a favorable safety profile. No significant cytotoxicity or *in vivo* toxicity was observed at effective concentrations, aside from mild spleen enlargement. Previous studies have demonstrated that VLX600 is less cytotoxic to normal cells compared to cancer cells (17). However, considering that the plasma concentrations required to achieve MIC levels for *M. abscessus* treatment exceed those for cancer therapy, a thorough evaluation of its therapeutic index is necessary.

Despite its promise, this study has some limitations. While this study identified iron chelation as the primary mechanism of VLX600, the specific molecular targets within *M. abscessus* remain unknown. Iron plays a central role in mycobacterial physiology, serving as a cofactor for enzymes and mediating electron transfer (44). Identifying the primary growth-limiting targets of VLX600 could enhance our understanding of its mechanism and inform the design of improved derivatives.

In conclusion, VLX600 is a promising antimicrobial agent with potential for clinical application in *M. abscessus* infections. Its iron-chelating mechanism, synergistic effects with standard antibiotics, and efficacy in preclinical models position it as a candidate for further development. Future research should focus on optimizing combination therapies, evaluating long-term safety, and identifying its specific targets to maximize its therapeutic potential.

## MATERIALS AND METHODS

### Chemical library

The chemical library consists of 3,200 chemical compounds that have been studied in clinical settings. These compounds were provided in solution in dimethyl sulfoxide (DMSO) at a concentration of 5 mM and stored at −80°C.

VLX600 used in further studies following the screening was purchased from MedChemExpress (New Jersey, USA).

### Bacterial strains and culture conditions

Type strains of *M. abscessus* subsp. *abscessus*, *massiliense*, and *bolletii* (ATCC19977, KCTC19086, and KCTC 19281, respectively) were used in this study.

Eighteen clinical isolates of *M. abscessus* were isolated from patients with a pulmonary *M. abscessus* infection and provided by Seoul National University Hospital.

Type strains of *M. avium, M. intracellulare* subsp. *yongonense, M. paraintracellulare, M. gordonae, M. paragordonae, M. marinum, M. vaccae, M. terrae, M. chleonae* subsp. *bovistauri, M. chelonae* subsp. *gwanakae* (ATCC 25291, 05-1390[T], MOTT64, ATCC14470, 49061[T], ATCC927, ATCC15483, ATCC15755, QIA-37, and MOTT36W, respectively), and the *M. bovis* BCG strain Tokyo were used in this study

All mycobacteria were cultured in complete 7H9 broth supplemented with 10% albumin-dextrose-catalase (ADC), 2.5% glycerol, and 0.2% Tween-80 until they reached the stationary phase in a shaking incubator at 37°C except for *M. marinum, M. gordonae,* and *M. paragordonae*, which were grown at 30°C. The addition of 0.2% Tween-80 was used to maintain a homogeneous culture (45, 46). For solid cultures, mycobacteria were grown on complete 7H10 plates supplemented with 10% oleic acid-albumin-dextrose-catalase (OADC) and 0.5% glycerol.

*E. coli, S. aureus,* and *P. aeruginosa* (NCCP 14541, NCCP 14780, and NCCP 14570, respectively) were kindly provided by the National Culture Collection for Pathogens (Chungju, Korea) and cultured in Luria-Bertani broth or on agar plates at 37°C.

## Construction of bioluminescent *M. abscessus* strains

pMV306hsp+luxG13 was a gift from Brian Robertson & Siouxsie Wiles (Addgene plasmid # 26161; https://www.addgene.org/26161/; RRID: Addgene_26161).

Three clinical isolates provided by Seoul National University were used to generate bioluminescent *M. abscessus* strains. Two isolates of *M. abscessus* subsp. *abscessus* with a rough morphotype and smooth morphotype and an isolate of *M. abscessus* subsp. *massiliense* with a rough morphotype were used.

Bacteria were grown in complete 7H9 broth until they reached $OD_{600}$ = 0.8–1.2. The samples were subsequently washed three times with 10% glycerol and electroporated with 1.5 µg of pMV306hsp+luxG13 using a Gene Pulser apparatus (Bio-Rad) at 2.5 kV, 25 µF, and 1,000 Ω followed by recovery in complete 7H9 broth at 37°C overnight.

For the isolation of bioluminescent bacteria, the suspension was spread on complete 7H10 plates containing 100 µg/mL kanamycin. Each colony picked from the selection plate was then cultured in complete 7H9 broth containing 100 µg/mL kanamycin, and the luminescence was confirmed using a Tecan F200 microplate reader.

## Neutral red uptake assay

J774A.1 cells were cultured in complete RPMI 1640 supplemented with 10% fetal bovine serum (FBS) and penicillin-streptomycin at 37°C in a humidified atmosphere with 5% $CO_2$. A total of $1 \times 10^5$ cells were seeded on a transparent 96-well cell culture plate and incubated overnight before drug application.

The drugs were diluted in RPMI 1640 supplemented with 2% FBS without penicillin-streptomycin and applied to the prepared J774A.1 cells for 48 h. The cells were subsequently stained with complete RPMI 1640 containing 40 µg/mL neutral red (Sigma-Aldrich, Seoul, Korea) at 37°C for 2 h. The cells were then washed three times with PBS followed by destaining with destaining buffer (50% ethanol, 49% distilled water, and 1% glacial acetic acid). The $OD_{540}$ was measured using a Tecan F200 microplate reader.

## *M. abscessus* infection assay in J774A.1 cells

J774A.1 cells were cultured in complete RPMI 1640 supplemented with 10% FBS and penicillin-streptomycin at 37°C in humidified atmosphere with 5% $CO_2$. A total of $1 \times 10^5$ cells were seeded on a 96-well cell culture plate and incubated overnight before the infection step.

*M. abscessus* was cultured in complete 7H9 broth until it reached the stationary phase before infection. The bacterial suspension was centrifuged (13,000 rpm, room temperature, 1 min) and then resuspended in PBS containing 1% Tween-80, followed by passing through a syringe multiple times to remove clumps. The suspensions were subsequently diluted in complete RPMI 1640 without penicillin-streptomycin and applied to J774A.1 cells at an MOI of 10 in a 96-well plate and incubated for 2 h. The infected cells were washed three times with PBS and treated with 50 µg/mL amikacin for 1 h to eliminate extracellular mycobacteria.

## MIC and MBC determination

The broth microdilution assay was performed according to CLSI guidelines. Bacteria were grown until they reached stationary phase. The bacterial suspension was centrifuged (13,000 rpm, room temperature, 1 min) and then resuspended in PBS containing 1% Tween-80, followed by passing through a syringe multiple times to remove clumps. The suspensions were subsequently diluted in CAMHB to McFarland standard 0.5 and further diluted 1:100 in CAMHB.

Next, 50 µL of the diluted bacterial suspension was seeded in a 96-well plate. Antibiotic agents, including VLX600, were serially diluted twofold, and 50 µL of each dilution was added to a 96-well plate containing the seeded bacteria. The plates were incubated at 37°C for 1–14 days, depending on the bacterial species, after which the $OD_{600}$ was measured. Wells containing only the drug were also prepared and used as a blank control. The inhibition rate of each well was calculated as follows:

$$\left(1 - \frac{\text{OD600 of the test well}}{\text{OD600 of non-treated well}}\right) \times 100\ (\%)$$

For slowly growing mycobacteria, such as *M. bovis* BCG, *M. avium*, *M. intracellulare*, *M. paraintracellulare*, *M. terrae*, *M. marinum*, *M. gordonae*, and *M. paragordonae*, 5% OADC was supplemented in CAMHB. The MICs were defined as the lowest concentrations of antibiotics that inhibited bacterial growth by at least 90%.

After the determination of the MIC in the broth microdilution assay, the bacterial suspensions from the wells treated with drug concentrations greater than the MIC were harvested. The suspensions were serially diluted in PBS and spread on complete 7H10 agar plates. The agar plates were incubated at 37°C until colonies appeared. The MBC was defined as the lowest concentration of antibiotics that killed at least 99.9% of the bacteria.

## CFU assay

The infected cells were washed three times with PBS and then treated with 1% Triton X-100 for 10 min to lyse the cells. Lysed cell suspensions were serially diluted in PBS, and the dilutions were spread on complete 7H10 agar plates. The agar plates were then incubated at 37°C until colonies appeared. The colonies were counted, and the actual CFUs were calculated by multiplying the number of counted colonies by the dilution factor.

## Checkerboard assay and synergy evaluation

For the direct treatment model, a type strain of *gordonae abscessus* subsp. *abscessus* was grown until it reached the stationary phase. The bacterial suspension was centrifuged (13,000 rpm, room temperature, 1 min) and then resuspended in PBS containing 1% Tween-80, followed by passing through a syringe multiple times to remove clumps. The suspension was subsequently diluted in CAMHB to McFarland standard 0.5 and further diluted 1:100 in CAMHB. Next, 100 µL of the diluted bacterial suspension was seeded in a transparent 96-well plate. VLX600 and antibiotics were serially diluted twofold into CAMHB. Fifty microliters of VLX600 was loaded into a 96-well plate in a gradient arrangement from top to bottom, with a low concentration at the top and a high

concentration at the bottom. Similarly, 50 µL of antibiotics was loaded into the same 96-well plate in a gradient arrangement from left to right, with a low concentration on the left and a high concentration on the right. The plates were incubated at 37°C for 1 week, and the $OD_{600}$ was measured using a Tecan F200 microplate reader. Synergy was evaluated with the FICI, which was calculated as follows:

$$\left(\frac{\text{MIC of drug A in combination with B}}{\text{MIC of drug A alone}}\right) + \left(\frac{\text{MIC of drug B in combination with A}}{\text{MIC of drug B alone}}\right)$$

The criteria are as follows: FICI ≤0.5 indicates synergy, 0.5 < FICI ≤ 4 indicates indifferent, and FICI >4 indicates antagonism.

For the J774A.1 infection model, J774A.1 cells were infected with bioluminescent *M. abscessus* (ID number 5) at an MOI of 10, as described previously. VLX600 and antibiotics were serially diluted twofold in RPMI 1640 supplemented with 2% FBS without penicillin-streptomycin. One hundred microliters of diluted VLX600 and antibiotics were added in the same manner as in the direct treatment model. The inhibition rate of each well was calculated as follows:

$$\left(1 - \frac{\text{RLU of the test well}}{\text{RLU of non-treated well}}\right) \times 100 \, (\%)$$

Synergy was evaluated by calculating the inhibition rate with four reference models, ZIP, Bliss, Loewe, and HSA, using the SynergyFinder web application. Synergy was evaluated with the synergy score as follows: a score ≥5 indicates synergy, −5 < score < 5 indicates an indifferent action, and a score ≤−5 indicates antagonism.

## Animal study

Seven-week-old female BALB/c (~20 g) mice were purchased from OrientBio (Seongnam, Korea). The mice were housed under specific pathogen-free conditions on a 12 h light and 12 h dark cycle at 18°C–23°C with 40%–60% humidity at the Seoul National University College of Medicine. The experiment was initiated after a 7-day acclimation period from the day the mice arrived.

A pulmonary *M. abscessus* infection model in mice was established by intraperito-neally injecting 150 mg/kg cyclophosphamide in a 0.2 mL volume on days 1 and 4 prior to infection. A clinical isolate (ID number 5) of *M. abscessus* with a rough morpho-type (1 × 10^6 CFUs in 40 µL) was prepared in PBS after clumps were removed and injected intranasally under anesthesia using 2%–5% isoflurane in a chamber on day 0. Starting the day after infection, the mice were divided into five groups (five mice per group) and received daily intraperitoneal injections of 0.1 mL of vehicle (PBS), 5 mg/kg VLX600, 10 mg/kg VLX600, 50 mg/kg amikacin, or a combination of 5 mg/kg VLX600 and 50 mg/kg amikacin. The mice were then euthanized by cervical dislocation on day 12 while under anesthesia.

The spleen was excised from the mice and weighed after gently blotting off excess water using a paper towel. The weight of the spleen was normalized to the body weight of each mouse.

The postcaval lobe of the lung was fixed with 10% formalin at 4°C for the histological analysis through H&E staining, while the other lobes were homogenized in PBS for the CFU assay. The homogenates were serially diluted in PBS and then spread on complete 7H10 agar plates to count the CFUs in the lungs. The agar plates were incubated at 37°C until colonies appeared. The colonies were counted, and the actual CFUs were calculated by multiplying the number of counted colonies by the dilution factor.

## ACKNOWLEDGMENTS

This research was supported by the National Research Foundation of Korea (NRF) grant funded by the Korea government (MSIT) (RS-2025-00553721) and was also supported by

a grant no. 16-2023-0011 from SNUBH Research Fund. The funders were not involved in the study design, data analysis, and writing of the manuscript.

All clinical isolates of *M. abscessus* were provided by Jae-Joon Yim and Nakwon Kwak from Seoul National University Hospital.

The chemical library used in this study was kindly provided by the Korea Chemical Bank (www.chembank.org) of the Korea Research Institute of Chemical Technology (Korea, Daejeon).

## AUTHOR AFFILIATIONS

[1]Department of Microbiology and Immunology, College of Medicine, Seoul National University, Seoul, Republic of Korea

[2]Department of Biomedical Sciences, College of Medicine, Seoul National University, Seoul, Republic of Korea

[3]Cancer Research Institute, College of Medicine, Seoul National University, Seoul, Republic of Korea

[4]Division of Pulmonary and Critical Care Medicine, Department of Internal Medicine, Seoul National University Hospital, Seoul, Republic of Korea

[5]Department of Internal Medicine, Seoul National University College of Medicine, Seoul, Republic of Korea

[6]Liver Research Institute, College of Medicine, Seoul National University, Seoul, Republic of Korea

[7]Institue of Endemic Diseases, Seoul National University, Seoul, Republic of Korea

## AUTHOR ORCIDs

Jaehun Oh  http://orcid.org/0009-0004-8659-8636
Nakwon Kwak  http://orcid.org/0000-0002-1897-946X
Bum-Joon Kim  http://orcid.org/0000-0003-0085-6709

## FUNDING

| Funder | Grant(s) | Author(s) |
| --- | --- | --- |
| National Research Foundation of Korea (NRF) | RS-2025-00553721 | Jaehun Oh |
|  |  | Seaone Choi |
|  |  | Hyejun Seo |
|  |  | Dong Hyun Kim |
|  |  | Hyelin Kim |
|  |  | Duhyung Lee |
|  |  | Junghwa Jang |
|  |  | Sangkwon Jung |
|  |  | Ju-Young Lee |
|  |  | Ziyun Kim |
|  |  | Bum-Joon Kim |
| SNUBH Research Fund | 16-2023-0011 | Jaehun Oh |
|  |  | Seaone Choi |
|  |  | Hyejun Seo |
|  |  | Dong Hyun Kim |
|  |  | Hyelin Kim |
|  |  | Duhyung Lee |
|  |  | Junghwa Jang |
|  |  | Sangkwon Jung |
|  |  | Ju-Young Lee |

| Funder | Grant(s) | Author(s) |
|---|---|---|
| | | Ziyun Kim |
| | | Bum-Joon Kim |

## AUTHOR CONTRIBUTIONS

Jaehun Oh, Conceptualization, Data curation, Formal analysis, Investigation, Methodology, Project administration, Visualization, Writing – original draft | Seaone Choi, Formal analysis, Investigation, Methodology, Visualization, Writing – original draft | Hyejun Seo, Formal analysis, Methodology | Dong Hyun Kim, Formal analysis, Methodology | Hyelin Kim, Formal analysis, Methodology | Junghwa Jang, Formal analysis, Methodology | Sangkwon Jung, Formal analysis, Methodology | Ju-Young Lee, Formal analysis, Methodology | Ziyun Kim, Formal analysis, Methodology | Jae-Joon Yim, Data curation, Formal analysis, Investigation, Supervision, Validation | Nakwon Kwak, Data curation, Formal analysis, Investigation, Supervision, Validation | Bum-Joon Kim, Conceptualization, Project administration, Supervision, Writing – original draft.

## DATA AVAILABILITY

All data generated or analyzed during this study are included in this published article and its supplemental material.

## ETHICS APPROVAL

The animal study was conducted in strict accordance with guidelines set forth by the Seoul National University Institutional Animal Care and Use Committee (IACUC). All procedures involving animals were approved by the IACUC (SNU-240228-3). The study adhered to all ethical principles and regulatory standards for the care and use of laboratory animals.

## ADDITIONAL FILES

The following material is available online.

### Supplemental Material

**Supplemental material (Spectrum00719-25-s0001.pdf).** Tables S1 and S2; Fig. S1 to S6.

### Open Peer Review

**PEER REVIEW HISTORY (review-history.pdf).** An accounting of the reviewer comments and feedback.

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
