## [Reviewer comments · Microbiology Spectrum]

Microbiology Spectrum

VLX600, an anticancer iron chelator, exerts antimicrobial effects on *Mycobacterium abscessus* infections

Jaehun Oh, Seaone Choi, Hyejun Seo, Dong Hyun Kim, Hyelin Kim, Duhjung Lee, Junghwa Jang, Sangkwon Jung, Ju-Young Lee, Ziyun Kim, Jae-Joon Yim, Nakwon Kwak, and Bum-Joon Kim

Corresponding Author(s): Bum-Joon Kim, Seoul National University College of Medicine

Review Timeline:

Submission Date:	March 12, 2025
Editorial Decision:	May 8, 2025
Revision Received:	May 11, 2025
Accepted:	May 15, 2025

Editor: Olivier Neyrolles

Reviewer(s): The reviewers have opted to remain anonymous.

Transaction Report:

DOI: <https://doi.org/10.1128/spectrum.00719-25>

Re: Spectrum00719-25 (VLX600, an anticancer iron chelator, exerts antimicrobial effects on Mycobacterium abscessus infections)

Dear Prof. Bum-Joon Kim:

Thank you for the privilege of reviewing your work. Below you will find my comments, instructions from the Spectrum editorial office, and the reviewer comments.

The revised manuscript was re-evaluated by one of the previous reviewers at AAC. The reviewer still raises concerns regarding Fig. 5C. This point is critical, as the data presented there are central to supporting the main claim stated in the title. We kindly request that you strengthen the explanation and provide a clearer defense of the interpretation of these results, as this is essential for the validity of the title's claim. Also, please provide the Y-axis on a logarithmic scale to better illustrate the data trends. In addition, including the raw data in a supplementary table would be very helpful for transparency and further assessment.

Revision Guidelines

Sincerely,
Olivier Neyrolles
Editor
Microbiology Spectrum

Reviewer #1 (Comments for the Author):

1. The efficacy assessment of VLX600 in mouse is a very important part of this study. The Y axis markings in figure 5C is rather confusing. It goes from 5×10^3 to 1×10^4 to 1.5×10^4 to 2×10^4 with equal spaces between them. I am not convinced if the data has been adequately represented. This point was made in the review of the original manuscript but there is no response to it in the revised manuscript and the 'response to reviewers'. Please note that this point is critical because the title of the manuscript asserts that 'VLX600 exerts antimicrobial effects on *M. abscessus* infections'. Because of this claim made in the title of the manuscript, this piece of data that shows lung CFU levels in response to treatments is the most important.

Dear Dr. Olivier Neyrolles

Editor of *Microbiology Spectrum*

May 12, 2025.

Response to reviewer comments

Reviewer #1 (Comments for the Author):

1. The efficacy assessment of VLX600 in mouse is a very important part of this study. The Y axis markings in figure 5C is rather confusing. It goes from 5×10^3 to 1×10^4 to 1.5×10^4 to 2×10^4 with equal spaces between them. I am not convinced if the data has been adequately represented. This point was made in the review of the original manuscript but there is no response to it in the revised manuscript and the 'response to reviewers'. Please note that this point is critical because the title of the manuscript asserts that 'VLX600 exerts antimicrobial effects on *M. abscessus* infections'. Because of this claim made in the title of the manuscript, this piece of data that shows lung CFU levels in response to treatments is the most important.

I apologized for not addressing to your feedback in the previous revision. I now fully understand the issue you pointed out regarding the Y axis markings in Fig. 5C. The Y axis has been changed to logarithmic scale to more accurately represent the distribution of CFU values, and the spacing between ticks has been adjusted accordingly. This change ensure that the data are properly visualized and interpreted. Thank you for your valuable suggestion.

Re: Spectrum00719-25R1 (VLX600, an anticancer iron chelator, exerts antimicrobial effects on Mycobacterium abscessus infections)

Dear Prof. Bum-Joon Kim:

Congratulations!

Your manuscript has been accepted, and I am forwarding it to the ASM production staff for publication. Your paper will first be checked to make sure all elements meet the technical requirements. ASM staff will contact you if anything needs to be revised before copyediting and production can begin. Otherwise, you will be notified when your proofs are ready to be viewed.

Sincerely,
Olivier Neyrolles
Editor
Microbiology Spectrum